# Oncology Therapeutics Targeting the Metabolism of Amino Acids

**DOI:** 10.3390/cells9081904

**Published:** 2020-08-15

**Authors:** Nefertiti Muhammad, Hyun Min Lee, Jiyeon Kim

**Affiliations:** Department of Biochemistry and Molecular Genetics, University of Illinois, Chicago, IL 60607, USA; nmuh@uic.edu (N.M.); hmlee19@uic.edu (H.M.L.)

**Keywords:** cancer metabolism, amino acids, oncogenic therapeutics

## Abstract

Amino acid metabolism promotes cancer cell proliferation and survival by supporting building block synthesis, producing reducing agents to mitigate oxidative stress, and generating immunosuppressive metabolites for immune evasion. Malignant cells rewire amino acid metabolism to maximize their access to nutrients. Amino acid transporter expression is upregulated to acquire amino acids from the extracellular environment. Under nutrient depleted conditions, macropinocytosis can be activated where proteins from the extracellular environment are engulfed and degraded into the constituent amino acids. The demand for non-essential amino acids (NEAAs) can be met through de novo synthesis pathways. Cancer cells can alter various signaling pathways to boost amino acid usage for the generation of nucleotides, reactive oxygen species (ROS) scavenging molecules, and oncometabolites. The importance of amino acid metabolism in cancer proliferation makes it a potential target for therapeutic intervention, including via small molecules and antibodies. In this review, we will delineate the targets related to amino acid metabolism and promising therapeutic approaches.

## 1. Introduction

Cancer cells satisfy the demand for rapid proliferation by increasing the nutrient flux through biosynthetic pathways. In many cancer cells, glucose is the most consumed metabolite. Otto Warburg observed an oxygen-independent preference for increased glycolysis in malignant cells [1]. While the majority of glucose is metabolized to and excreted as lactate, some can be used in adjacent metabolic routes such as the pentose phosphate pathway (PPP), hexosamine biosynthesis pathway, and NEAA and nucleotide synthesis pathways. In addition to glucose, cancer cells rely on amino acid metabolism for bioenergetic, biosynthetic, and redox balance support.

Glutamine is essential for most cancer cell growth. Some cancer cells cannot survive in the absence of exogenous glutamine and show glutamine addiction [2]. Through glutaminolysis, glutamine carbons can be fed into the tricarboxylic acid (TCA) cycle, where glutamine is converted to glutamate and subsequently α-ketoglutarate (α-KG). Glutamine is also key to nitrogen metabolism in the cell [2]. The amide (γ) nitrogen directly contributes to nucleobase synthesis and hexosamine biosynthesis whereas the amine (α) nitrogen is important for transaminase reactions. Following conversion to glutamate, the remaining α-nitrogen from glutamine can be reversibly transferred to ketoacids to generate NEAAs.

Serine is the third most consumed metabolite in cancer cells [3] and contributes to nucleotide synthesis through folate-mediated one carbon metabolism [4]. Serine metabolism is also crucial to redox balance. Although most reducing equivalents are produced from the PPP, serine metabolism provides another major source [5]. Along with serine, cysteine and methionine are also involved in redox balance. The methionine cycle participates in redox control through the glutathione (GSH) synthesis. GSH is an important ROS scavenger and the methionine cycle is connected to GSH production through the transsulfuration pathway, a metabolic pathway interconverting cysteine and homocysteine through the intermediate cystathionine. Cysteine, the key element to generate GSH, can be synthesized by the transsulfuration pathway, but is also taken up by cells through the xCT transporter as cystine, the oxidized cysteine dimer. Due to the high demand of cysteine for ROS detoxification in cancer cells, xCT upregulation often accompanies malignant transformation. Methionine metabolism is also critical for epigenetic remodeling in cancer. The methionine cycle facilitates methylation-modification reactions through generation of S-adenosylmethionine (SAM).

Evading immune control is also essential to cancer survival. Kynurenine, an oncometabolite produced from the tryptophan catabolism, can suppress T-cell differentiation [6]. The transcellular activity of kynurenine can alter the transcriptional activity of immune cells to evade detection [6]. The polycationic alkylamine metabolites called polyamines are also downstream products of amino acid catabolism, specifically of arginine. Polyamines also function as oncometabolites and their metabolism is often dysregulated in cancer. Polyamine metabolism is important for oxidative protection, cell-to-cell communication, and the synthesis of proteins and nucleic acids [7,8].

The functional roles of amino acids in cancer have been extensively discussed [9,10]. In this review, we examine how cancer cells acquire and use amino acids. Cancer-associated amino acid transporters, transaminases, and macropinocytosis, an alternative way to obtain proteins, are explained. We highlight three components regarding the usage of amino acids: (1) Biomass-related nucleotide synthesis, (2) signaling-related methylation and amino acid-derived oncometabolites, and (3) redox balance-related GSH synthesis. Further, we discuss what amino acid metabolism-associated vulnerabilities can be targeted in cancers and what progress has been made in drug discovery and preclinical studies. Finally, the Food and Drug Administration (FDA) approved therapeutic interventions and those that have entered clinical trials are described.

## 2. Amino Acid Transporters

Amino acids are trafficked across organelles and the plasma membrane by means of membrane-bound amino acid transporters (AATs). Transport by AATs is often coupled to the movement of ions (e.g., Na^+^, H^+^, K^+^, Cl^−^) to permit amino acid transfer against electrochemical concentration gradients. Moreover, molecular recognition by AATs is most often based on the chemical properties (e.g., hydrophilicity, net electrical charge) of the amino acid rather than the shape. Thus, a single AAT can facilitate the transport of multiple amino acids and enable transport redundancy where needed [11,12]. To date, approximately 60 AATs have been classified based on substrate specificity and transport mechanism (e.g., antiporter, coupling ions including Na^+^, H^+^, K^+^) [11]. Altered expression and transport behavior is associated with cancer in about a dozen AATs.

Among AATs upregulated in cancer, the alanine-serine-cysteine transporters 2 (ASCT2, encoded by *SLC1A5*) has been spotlighted as a therapeutic target because it is the primary glutamine transporter [13] and many cancers show glutamine addiction [2]. ASCT2 is induced by the c-Myc transcription factor, and is highly expressed in various cancers including colorectal, prostate, lung, and breast cancer [14,15,16,17]. In recent years, therapeutic targeting of ASCT2 has led to the development of antibody and small molecule inhibitors. KM8094 is an ASCT2 specific monoclonal antibody that shows antitumor effects in a patient-derived xenograft (PDX) mouse model of gastric cancer [18], supporting further progression of KM8094 into future clinical trials. MEDI7247 is a novel pyrrolobenzodiazepine antibody-drug conjugate targeting ASCT2 (Figure 1A). It shows potent in vivo antitumor activity in a variety of hematological cancers and solid tumors and has been evaluated in a Phase 1 clinical trial in hematological cancer (NCT03106428) (Table 1) [19].

Along with antibodies, several promising small molecule inhibitors of ASCT2 have been developed. V-9302 (2-amino-4-bis(aryloxybenzyl)aminobutanoic acid) is a competitive small molecule antagonist of transmembrane glutamine flux and has been reported to selectively target ASCT2. The V-9302 treatment attenuates cancer cell growth both in vitro and in vivo, implying potential therapeutic benefits of V-9302 for cancer patients with high ASCT2 [20]. Benzylserine and benzylcysteine are competitive inhibitors of ASCT2 [21] and have been reported to suppress cancer cell growth (benzylserine in breast cancer cells and benzylcysteine in gastric cancer cells, respectively) [22,23]. L-γ-Glutamyl-p-nitroanilide (GPNA) also targets ASCT2. It is a structural analog of glutamine that inhibits Na^+^-dependent AATs including ASCT2. Numerous studies have shown that GPNA treatment reduces glutamine uptake and cell viability, mainly through mechanistic target of rapamycin kinase complex 1 (mTORC1) signaling inhibition [17,24,25,26]. While promising, the described ASCT2 inhibitors need additional optimization to enable future development. Dose regimens need further evaluation for benzylserine and benzylcysteine, as millimolar concentrations are required to block cancer cell growth even in vitro [22]. Since Na^+^-dependent AATs are not limited to ASCT2, GPNA can inhibit other transporters such as L-type amino acid transporter 1 (LAT1, encoded by *SLC7A5*) and System A transporters SNAT1 (encoded by *SLC38A1*) and SNAT2 (encoded by *SLC38A2*). Potential off-target effects of GPNA should be cautiously examined.

ASCT2-mediated glutamine influx is coupled with LAT1. LAT1 has high affinity for the transport of branched chain amino acids (BCAAs; Val, Leu, and Ile) and bulky amino acids (Phe, Tyr, Trp, His, and Met). Similar to ASCT2, LAT1 is transactivated by c-Myc [27] and is highly expressed in various cancers including breast [28] and lung cancer [29,30]. Among LAT1 inhibitors, a tyrosine analogue called KYT-0353 (JPH203) has shown therapeutic efficacy [31]. It was designed based on the structure of the thyroid hormone triiodothyronine (T3), a known LAT1 inhibitor [32]. JPH203 is a non-transportable blocker and inhibits LAT1 in a competitive manner with a K_i_ value of 2.1 μM (Figure 1A). It induced cytostatic growth arrest in genetically engineered mouse models (GEMMs) of anaplastic thyroid carcinoma and gastric cancer [33] and inhibited in vitro cancer cell growth including oral cancer [34] and osteosarcoma [35]. JPH203 also showed promising results in a Phase 1 trial [31] where six out of seventeen patients with advanced solid tumors (colorectal, breast, biliary tract, pancreas, and esophagus cancer) had a partial response or stable disease. Four out of the six responders were diagnosed with biliary tract cancer where plasma levels of LAT1 substrates remained high after the JPH203 treatment [31]. This result suggests that high plasma levels of LAT1 substrates could serve as predictive biomarker of JPH203 efficacy, at least in biliary tract cancer. JPH203 is also being evaluated in a Phase 2 clinical trial in patients with advanced biliary tract cancers (Japan UMIN Clinical Trials Registry UMIN000034080) (Table 1). Another tyrosine analogue LAT1 inhibitor, SKN103, also suppresses cancer cell growth [32]. Importantly, SKN103 additively reduces cancer cell growth when combined with cisplatin [32]. While it has yet to be tested in vivo, SKN103 could potentially enhance the therapeutic efficacy of chemotherapies in patients. Another promising therapeutic, 2-aminobicyclo [2.2.1]heptane-2-carboxylic acid (BCH), is a competitive inhibitor of LAT1. It has been shown to reduce cancer cell growth in vitro, including breast, prostate, and lung [36,37,38], and in vivo in cholangiocarcinoma and esophageal squamous carcinoma xenograft [39,40].

Cystine/glutamate antiporter xCT (encoded by *SLC7A11*) imports cystine into cells while exporting out glutamate. xCT functionally couples to ASCT2 in cancer because ASCT2 can functionally provide glutamate by importing glutamine, which is converted to glutamate by a glutaminase (GLS) reaction [41]. Inside the cell, cystine is reduced to cysteine, a rate-limiting precursor of the reduced form of glutathione (GSH), a ROS scavenger. xCT activity can thereby help cancer cells mitigate ROS toxicity. xCT upregulation is known to lower the efficacy of ROS-inducing chemotherapeutic drugs (e.g., geldanamycin) [42,43], making xCT a worthwhile therapeutic target. Several therapeutic drugs, originally designed to target unassociated pathways, have been reported to inhibit xCT. Sulfasalazine, an anti-inflammatory treatment approved by the FDA, appears to suppress xCT-mediated uptake of cystine [44] and attenuate growth of xenografted tumors derived from breast cancer cells [41], non-small cell lung cancer (NSCLC) cells [45], and lymphoma cells [46] (Figure 1A and Table 2). Erastin is a small molecule that initiates ferroptosis [47]. Initially, Erastin was identified as a direct inhibitor of voltage-dependent anion channel (VDAC)2 and VDAC3 [48]. However, it was later found to functionally inhibit xCT [49]. Erastin treatment increases ROS and shows decreased survival of cells derived from sarcoma [50], fibrosarcoma [51], and lung cancer [52], but it has yet to be validated whether the antitumor effect of erastin is directly linked to xCT inhibition. Sorafenib is another inhibitor that can functionally block xCT activity (Figure 1A). It is an FDA-approved multi-receptor tyrosine kinase (RTK) inhibitor used in a number of cancers including liver [53], kidney [54], and thyroid cancer [55] (Table 2). Similar to erastin, it was shown to inhibit xCT function and trigger ferroptosis [52]. Since erastin or sorafenib-mediated cancer cell death is partly related to excessive ROS, treatment with these drugs alongside other known anti-cancer drugs (e.g., cisplatin and geldanamycin) would likely increase antitumor effects. While these functional xCT inhibitors effectively block cystine uptake and have been widely used in preclinical studies as xCT inhibitors, one should evaluate possible off-target effects and carefully examine whether cancer cell death mediated by these medications is associated with xCT inhibition alone. In addition to targeting xCT [56], sulfasalazine also inhibits immunity-related signaling pathways such as NF-κB [57] and redox pathway proteins such as sepiapterin reductase [58] and reduced folate carrier [59]. Developing specific xCT inhibitors would provide a means to minimize off-target effects and directly sensitize tumors to a ROS-inducing drug treatment.

## 3. Macropinocytosis, an Alternative Way to Obtain Amino Acids

Obtaining nutrients from the environment is key to the sustainable and rapid growth of cancer cells. This is especially true of cancers in nutrient-depleted microenvironments such as those characterized by restricted blood flow and low oxygen levels. To circumvent these metabolic obstacles, cancer cells utilize alternative mechanisms to acquire nutrients. Among the alternative mechanisms, macropinocytosis is the best-known process to obtain exogenous amino acids. Macropinocytosis is an endocytic process where extracellular macromolecules, such as serum albumin, are engulfed in vesicles called macropinosomes [60]. Once internalized by macropinocytosis, proteins can be degraded to amino acids in the lysosome. This allows cancer cells to survive and proliferate under the nutrient-poor conditions of the tumor microenvironment [61,62]. Macropinocytosis can be induced by oncogenic transformation (e.g., oncogenic RAS or activated v-Src expression) or loss of function mutation in tumor suppressors (e.g., loss of phosphatase and tensin homolog (PTEN)) [63,64,65]. Thus, it is highly active in tumors with activated RAS such as pancreatic, lung, and colon cancer [61,66,67]. In this context, macropinocytosis can confer metabolic liabilities. The most commonly used preclinical inhibitors for macropinocytosis are the Na^+^/H^+^ exchanger (NHE) inhibitors amiloride and 5-[N-ethyl-N-isopropyl] amiloride (EIPA). The mechanism of action of amiloride and EIPA as macropinocytosis inhibitors is related to actin polymerization. NHE promotes actin polymerization mediated by Rac1 and Cdc42 [68], which induces membrane ruffling, a critical early step leading to the initiation of macropinocytosis (Figure 1B). Oncogenic Ras activates Rac1 and Cdc42 [69,70], which in turn stimulates p21-activate kinase 1 (PAK1) to induce actin polymerization [71,72] whereas PI3Ks cooperate with Rac1 to regulate cell polarization in response to different stimuli [73]. EIPA and amiloride have been shown to inhibit tumor growth in multiple mouse tumor models including prostate cancer [74] and pancreatic ductal adenocarcinoma (PDAC) [61]. Similar to the mechanism of amiloride and EIPA, FRAX597, a small molecule targeting PAK1, has also shown the ability to block macropinocytosis in *Pten* KO mouse embryonic fibroblast and induce cell death [74]. Two PI3K inhibitors in clinical trials, BKM120 [75] and ZSTK474, and an FDA-approved drug BYL719 [74], have been reported to suppress macropinocytosis (Figure 1B). BKM120 completed a Phase 3 clinical trial for breast cancer (NCT01610284) and a Phase 2 trial for lymphoma (NCT02301364) and lung cancer (NCT01297491) while ZSTK474 has been tested in a Phase 1 for advanced solid tumors (NCT01280487) (Table 1). It would be interesting to examine whether combining these drugs with current therapeutic regimens is beneficial for patients with highly macropinocytic tumors (e.g., RAS-activated tumors). Interestingly, small scale screening using 640 FDA-approved compounds has identified an antidepressant, imipramine, as a novel macropinocytosis inhibitor [76] (Figure 1B and Table 2). Similar to EIPA, imipramine inhibits membrane ruffle formation. It has inhibited macropinocytosis in several cell types including cancer cells, dendritic cells, and macrophages [76]. Given the lack of macropinocytosis inhibitors suitable for clinical use, imipramine could become a promising therapeutic drug once the anticancer effects are fully evaluated.

## 4. Transaminase, a Key Mechanism of NEAA Synthesis

While essential amino acids (EAAs) must be obtained from diet and taken up by amino acid transporters, NEAA can be synthesized endogenously. Most NEAAs are synthesized from glucose; either glycolytic intermediates (e.g., Ser, Gly, Ala) or TCA cycle intermediates (e.g., Asp, Asn, Glu) provide the carbon skeleton of NEAAs and the α-amino group can be obtained from preexisting amino acids (in most cases, glutamate) mediated by transaminases. Transaminases or aminotransferases are a group of enzymes that catalyze the reversible transfer of an α-amino group from an amino acid to an α-ketoacid. There are three main transaminases involved in NEAA synthesis. Aspartate transaminase (AST, also known as glutamic-oxaloacetic transaminase (GOT), and numbered 1 for the cytosolic form and 2 for the mitochondrial form), catalyzes reversible transfer of an α-amino group of glutamate to oxaloacetate, thus forming α-KG and aspartate. GOT1 is particularly important for redox balance and growth of PDAC [77]. Unlike most cells which utilize mitochondrial glutamate dehydrogenase (GDH) to convert glutamine-derived glutamate into α-KG to fuel the TCA cycle, PDAC cells transport glutamine-derived aspartate to the cytoplasm where it can be converted into oxaloacetate by GOT1. In the cytoplasm, conversion of oxaloacetate into malate and then pyruvate by the malic enzyme produces one equivalent of nicotinamide adenine dinucleotide phosphate (NADPH), subsequently increasing the NADPH/NADP^+^ ratio which can potentially maintain the cellular redox state [77]. Alanine transaminase (ALT, also known as alanine aminotransferase (ALAT)) catalyzes reversible conversion of glutamate to α-KG and pyruvate to alanine. Inhibition of ALT induces oxidative phosphorylation and subsequent increase of mitochondrial ROS, suggesting ALT as a potential target to promote oxidative stress and inhibit cancer cell growth [78]. Phosphoserine aminotransferase 1 (PSAT1) is the transaminase for serine. It transfers an α-amino group of glutamate to phosphohydroxypyruvate (PHP), a metabolite generated from glycolytic intermediate 3-phosphoglycerate (3PG) by phosphoglycerate dehydrogenase (PHGDH). PSAT1 expression is elevated in colon cancer, esophageal squamous cell carcinoma (ESCC) and NSCLC, and has been shown to enhance tumor growth, metastasis, and chemoresistance [79,80,81,82]. BCAAs need to be obtained from outside the cells via transporters because they are EAAs. However, cells can technically synthesize BCAAs if branched chain keto-acids (BCKAs) are available. Branched chain amino acid aminotransferase (BCAT, 1 for cytosolic form and 2 for mitochondrial form) catalyzes reversible transfer of an α-amino group of isoleucine, leucine, or valine to α-KG, thus forming glutamate and α-keto-β-methylvalerate, α-ketoisocaproate, or α-ketoisovalerate. In cancers, BCATs enhance BCAA uptake to sustain BCAA catabolism, rather than BCKA to BCAA conversion, and support mitochondrial respiration [83,84]. Of the two isoforms, BCAT1 is the major enzyme implicated in cancer growth and is highly expressed in various cancers including glioblastoma (GBM) and ovarian cancer [85,86].

Due to cancer cells’ increased transaminase expression and the metabolic liabilities resulting from transaminase inhibition compared with normal tissues, transaminases have been suggested as an attractive target to selectively kill cancer cells. Among various amino acid transaminases, the drug discovery field has shone a spotlight on GOT1. However, the development of GOT1 inhibitors has been challenging, and hardly any compound has yet demonstrated selectivity for GOT1-dependent cell metabolism. Most inhibitors showing suppressive effects against GOT1 are either initial hit compounds from high throughput screening or compounds known to target other proteins. Aspulvinone O (AO), a natural product originating from the soil fungus *Aspergillus terreus*, is an initial hit compound that was recently identified as a GOT1 inhibitor from a natural product library screening effort [87]. Mechanistically, AO competitively binds to the activation site of GOT1, suppresses glutamine metabolism and reduces PDAC growth both in vitro and in vivo [87]. The lairson group has identified 4-(1H-indol-4-yl)-N-phenylpiperazine-1-carboxamide as a GOT1 inhibitor by high throughput screening [88]. After optimizing the initial GOT1 inhibitor, mouse pharmacokinetics studies implicated potency issues for the series. Further optimization identified multiple derivatives with >10-fold improvements in potency, suggesting that the improved GOT1 inhibitors could potentially be validated in cells and rodent xenograft models. PF-04859989 is a known kynurenine aminotransferase 2 (KAT2) inhibitor but has been shown to also inhibit GOT1 [89]. Although PF-04859989 has poor pharmacokinetic properties due to rapid O-glucuronidation of the hydroxamate group [90], further optimization (e.g., replacement of the hydroxylamine motif) could improve the pharmacokinetic profile and make the compound suitable for testing in vivo efficacy.

## 5. Serine and the One Carbon Metabolism

### 5.1. Serine De Novo Synthesis

Glutamine has long been appreciated as a critical NEAA for cancer cell proliferation and survival [2]. Recent reports have, however, uncovered the importance of other NEAAs in cancer cell viability and growth. Among them, serine stands out, supporting a number of metabolic processes that are required for cancer cell proliferation including glycine synthesis, nucleotide synthesis, DNA and histone methylation, and NADPH production, all of which are linked to one carbon (1C) metabolism (to be explained later).

Cellular access to serine is facilitated by several amino acid transporters or through the de novo synthesis. Serine biosynthesis begins with glycolysis. As an allosteric activator of pyruvate kinase isoform m2 (PKM2), serine regulates glycolytic flux [91]. When serine is depleted, the PKM2 activity is reduced and accumulated glycolytic intermediates are shunted into the serine de novo biosynthesis [91]. PHGDH catalyzes the first step of the serine biosynthesis pathway, oxidizing a glycolytic intermediate 3PG into PHP (Figure 2A). Then, the α-nitrogen is supplied by a PSAT1-catalyzed transaminase reaction to deliver PHP. Due to its central biological importance, cancer cells often have high demand for serine and thus activate serine biosynthesis mainly by upregulating PHGDH expression. A high PHGDH expression is observed in aggressive subtypes of cancers. It is upregulated in triple-negative breast cancers (TNBCs), estrogen receptor (ER)-negative breast cancers, metastatic variants of ER–negative breast cancer cells, gliomas, and cervical cancer. Importantly, its high expression in these tumors is associated with poor prognosis [81,92,93,94]. To date, there are no FDA-approved drugs against PHGDH. However, experimental inhibitors are being improved upon for future clinical testing. Some of the compounds identified as PHGDH inhibitors show an allosteric effect, disrupting oligomerization or conformational states. Thus, allosteric inhibition may be a promising strategy to block the PHGDH activity. CBR-5884 was identified as a PHGDH inhibitor from a library screening of 800,000 small drug molecules [95]. It inhibits PHGDH oligomerization and has a significant inhibitory effect in melanoma and breast cancer cell lines with a high PHGDH, especially when extracellular serine is absent [95]. CBR-5884, however, needs medicinal chemistry-based optimization to move into preclinical and clinical testing. It is unstable in mouse plasma and shows low solubility in vivo. NCT-502 and -503 are non-competitive molecules for PHGDH that impose reversible inhibition and destabilization. They were selected from the Molecular Libraries Small Molecule Repository (MLSMR) library of 400,000 compounds and demonstrated specificity against PHGDH over other dehydrogenases [96]. NCT-502/-503 have been shown to reduce serine biosynthesis and impair PHGDH-dependent cancer growth both in vitro and in vivo (Figure 2B). Both drugs have good potency (NCT-502: IC_50_ 3.7 ± 1 µM and NCT-503: IC_50_ 2.5 ± 0.6 µM) and reasonable ADME (absorption, distribution, metabolism, and excretion), suggesting their potential to move forward to clinical testing. More importantly, a recent study showed that NCT-503 works synergistically with sorafenib to inhibit hepatocellular carcinoma (HCC) growth in vivo [97]. The data imply that the combination treatment of NCT-503 and various tyrosine kinase inhibitors (TKIs) can be potentially used to overcome TKI drug resistance in HCC. PKUMDL-WQ-2101 and PKUMDL-WQ-2201 are allosteric inhibitors of PHGDH identified using a structure-based approach. Inhibition is achieved by stabilizing PHGDH in the inactive conformation, preventing the inactive sites from closing [98]. Both inhibitors demonstrated selectivity for PHGDH, reduced de novo serine biosynthesis in PHGDH-amplified cell lines, and decreased growth of tumors with a high PHGDH expression.

### 5.2. Folate Metabolism Pathway

Serine, if not used for protein synthesis, is shuttled into the tetrahydrofolate (THF)-mediated 1C metabolism pathway. THF-mediated 1C metabolism is a metabolic process providing 1C units for numerous biosynthesis pathways (e.g., DNA, polyamines, phospholipids), the methylation reaction and redox maintenance. THF is synthesized from dietary folate by dihydrofolate reductase (DHFR) (Figure 2A). It serves as a universal cofactor that chemically activates and carries 1C units on the N5 and/or N10 of THF at the oxidation states of formate (e.g., 10-formylTHF), formaldehyde (e.g., 5,10-methyleneTHF), or methanol (e.g., 5-methylTHF) (Figure 2A) [99]. THF-mediated 1C metabolism pathway is composed of two interlinking metabolic cycles—the folate and methionine cycles—and is compartmentalized to the cytoplasm, mitochondria, and nucleus [100] (Figure 2A). Serine is a critical amino acid in the folate cycle due to its role providing carbons for nucleotide synthesis and serving as a precursor for glycine production (Figure 2A). Enzymes participating serine-folate metabolism exist both in the cytoplasm and the mitochondria. Serine hydroxymethyltransferase (SHMT) interconverts serine and glycine. In the mitochondria, serine is preferentially catabolized to glycine by SHMT2 and releases a 1C unit, which is transferred to THF to generate 5,10-methyleneTHF. 5,10-methyleneTHF then transfers the 1C unit to deoxyuridine monophosphate (dUMP) to make deoxythymidine monophosphate (dTMP) by thymidylate synthase (TS). Glycine generated from SHMT2 reaction is also utilized for many biosynthetic pathways including glutathione synthesis, and as carbon backbone and nitrogen donors for purines. In the cytoplasm, serine is normally synthesized from glycine by SHMT1 but can be catabolized in the reverse reaction under serine deprived conditions (Figure 2A). The importance of 1C metabolism in cancer was initially recognized back in 1948 by Sidney Farber. After his finding that selective targeting of folate metabolism in leukemia patients with the folate antagonist aminopterin led a temporary remission [101], a series of THF analogues known as antifolates were developed. These drugs bind to and inhibit DHFR, thus blocking THF formation and 1C metabolism [102]. Methotrexate is the best known antifolate (Figure 2B and Table 2). It is an FDA-approved chemotherapy drug used in many types of cancers including leukemia and breast cancer [103,104]. However, it is difficult for methotrexate to cross the blood-brain barrier, resulting in extremely high dose administration in primary central nervous system lymphoma (PCNSL) [105]. A multi-targeted antifolate, pemetrexed, was approved for treatment against NSCLC and pleural mesothelioma (Table 2). Along with DHFR, pemetrexed targets the nucleotide biosynthesis enzymes including TS, glycinamide ribonucleotide formyltransferase (GART), and 5-aminoimidazole-4-carboxamide ribonucleotide transformylase (AICART) [106,107] (Figure 2B and Table 2). Molecular docking studies and binding assays indicate that SHMT1 is the target of classical antifolates [108]. Lometrexol, an antifolate and the first GART inhibitor to be investigated clinically [109,110], has also been shown to inhibit SHMT2 [111] (Figure 2B). Lometrexol has undergone a Phase 2 clinical trial (NCT00033722) (Table 1). Its safety and impact on patient survival were accessed in treating patients with previously treated stage IIIB or stage IV NSCLC. Lometrexol is also part of a Phase 1 trial in combination with paclitaxel to access the maximum tolerated dose and antitumor activity in patients with locally advanced or metastatic solid tumors (NCT00024310). To offset cytotoxicity, treatment with THF analogs are supplemented with folate.

While antifolates are widely used as a therapeutic treatment, they target multiple folate-dependent enzymes. Small molecule inhibitors could offer a target-specific therapy. SHIN1 is a competitive inhibitor of SHMT1 and 2 [99] with a median IC_50_ of 4µM across 300 human cancer cell lines. Although it binds both the SHMT1 and 2 active site, SHIN1 is more potent against cytosolic SHMT1 than mitochondrial SHMT2 [99]. Poor drug penetration due to the compartmentalization of SHMT2 into the mitochondria organelle could explain the asymmetrical effect of SHIN1. A greater overall enzymatic activity of SHMT2 due to high expression is another possibility. SHIN1 can deplete metabolites within and downstream of 1C metabolism and perturb the adjacent metabolism. In colorectal carcinoma cells, treatment with SHIN1 resulted in glycine, GSH and ADP depletion, and purine salvage intermediate accumulation [99]. While SHIN1 is suitable for cell culture studies, it is not stable in vivo due to rapid clearance [99]. Issues with a high clearance were addressed with the dual SHMT inhibitor SHIN2 [111]. SHIN2 is the first SHMT1/2 inhibitor to demonstrate in vivo efficacy and synergizes with FDA approved antifolate methotrexate in mouse primary T cell acute lymphoblastic leukemia (T-ALL) and in PDXes (Figure 2B). Although SHIN2 needs further validation, these promising outcomes may extend to more advanced preclinical models and clinical testing with other antitumor drugs in the future.

## 6. Methionine Metabolism

### 6.1. Methionine Cycle

In comparison to normal tissues, cancer cells are widely dependent on exogenous methionine for survival. Proliferation is impaired under methionine-depleting conditions partly due to cell cycle arrest for many cancer cells including lung adenocarcinoma, fibrosarcoma, and kidney carcinoma [112]. In addition, leukemia cells are unable to proliferate under methionine-depleted conditions even with the supplementation of homocysteine, a metabolite required for methionine synthesis by methionine synthase (MS) [113]. Methionine is metabolized through the methionine cycle, which is linked to nucleotide biosynthesis, GSH synthesis, and nucleotide/histone methyltransferase reactions. Enhanced utilization of pathways adjacent to the methionine cycle to meet the demands of rapid biosynthesis may explain methionine addiction in cancer cells. Cancer therapeutics related to methionine metabolism have been used in clinical trials and focus on methionine depletion, thus affecting those cancers exhibiting methionine addiction [114,115]. L-methionine-α-amino-γ-mercaptoethane lyase or methioninase (METase) is an enzyme that converts methionine to α-ketobutyrate, methanethiol, and ammonia [114,115]. METase is being explored as a therapeutic and has shown efficacy in melanoma and Ewing sarcoma PDXes, and orthotopic models of osteosarcoma [116,117,118]. A pilot Phase 1 clinical trial using recombinant METase was conducted in patients with advanced lymphoma, breast, lung, and renal cancer [119], showing methionine depletion without toxicity. The methionine cycle is tightly interwoven with the folate cycle in 1C metabolism (Figure 2A). The methyl donor SAM is generated from methionine and ATP by methionine adenosyltransferase (MAT), in the methionine cycle [120]. SAM is the methyl donor for all methylation reactions, including methylation of DNA, RNA, and histones. SAM is used by methyltransferases (MTs) and is essential to a vast number of cellular reactions [121]. Studies indicate that methionine addiction is actually an addiction to SAM. MTs generate S-adenosyl-homocysteine (SAH) by removing a methyl group from SAM. SAH is dephosphorylated into homocysteine by S-adenosyl-L-homocysteine hydrolase (SAHH) and MS uses 5-methylTHF to regenerate methionine through methylation.

### 6.2. SAM and Epigenetic Regulation

Epigenetic dysregulation, a feature pervasive in cancer tissues [122,123,124,125], can be linked to methionine dependence through SAM. As a survival response to hostile environments such as hypoxia, some cancer cells manipulate their epigenetic landscape [126]. Epigenetic dysregulation can occur due to DNA hypo- or hypermethylation, the former being more global and the latter more local in cancer tissues. DNA hypermethylation can have a pro-tumorigenic role, silencing tumor suppressors and conferring a growth advantage to malignant cells [126,127]. DNA hypomethylation can activate oncogenic genes to support malignant cells [128,129]. Nicotinamide N-methyltransferase (NNMT) uses SAM to methylate nicotinamide. NNMT is important for cancer cell migration, invasion, and proliferation, and is overexpressed in a variety of cancers, including lung, liver, bladder, and colon [130,131,132,133]. Depletion of SAM induces a hypomethylated state in cancer where pro-tumorigenic genes can be expressed [129]. Reversing DNA hypomethylation through SAM supplementation inhibited growth of gastric, colon, and liver cancer cells [134,135]. Since SAM supplementation had no adverse effect on tumor suppressor expression in normal tissues, SAM could be used as a cancer therapeutic [134]. SAM depletion, by contrast, can induce cell cycle arrest [136]. SAM levels may be a part of a G1 cell cycle check point that maintains an epigenetic balance of the cells.

### 6.3. Methionine Salvage Pathway

The methionine salvage pathway, also known as the 5′-methylthioadenosine (MTA) cycle, is another mechanism that can alter cellular methylation through SAM, the final product of the pathway [120]. Methionine is regenerated from the polyamine metabolism byproduct MTA. This sulfur-containing nucleoside undergoes six enzymatic steps to yield the deaminated precursor of methionine, 4-methylthio-2-oxobutyrate. Amino acid transaminases then carryout the amination step to yield methionine [137] which can be converted into SAM by MAT. Interestingly, the enzyme 1,2-dihydroxy-3-keto-5-methylthiopentene dioxygenase (ADI1), which catalyzes the penultimate step of the MTA cycle to form 4-methylthio-2-oxobutyrate keto-acid, has been implicated as a tumor suppressor [138]. ADI1 expression is inversely correlated with poor prognosis [138] and loss of ADI1 is observed in several cancer types including prostate and liver [138,139]. ADI1 accelerates the MTA cycle to elevate SAM levels, altering promoter methylation profiles and inhibiting hepatoma growth [138]. In addition to SAM and methionine levels, proteins in the methionine salvage pathway have also been identified as therapeutic targets. Loss of methylthioadenosine phosphorylase (MTAP), the first enzyme in the salvage pathway, is often observed in cancers including NSCLC [140], leukemia [141], and GBM [142]. MTAP loss often occurs with p16^INK4a^/CDKN2A deletion, which is prevalent in cancer [143]. Methionine adenosyltransferase 2A (MAT2A) is an enzyme that generates SAM from methionine and is identified as a metabolic vulnerability within MTAP-deleted tumors [143]. Targeting the methionine salvage and SAM biosynthesis pathways in the cells opens the door to small molecule therapeutics. AG-270 is a promising MAT2A inhibitor. A Phase I clinical trial was initiated to evaluate the activity, maximum tolerated dose, and pharmacokinetics of AG-270 in patients with advanced solid tumors or lymphoma with homozygous MTAP deletion (NCT03435250) (Figure 2B and Table 1). AG-270 is being evaluated as a single agent and in combination with taxane-based chemotherapies. PF-9366 is a potent and specific allosteric inhibitor of MAT2A and shares an overlapping binding site with the MAT2A regulator MAT2B. Binding of PF-9366 increases substrate affinity, but decreases enzyme turnover by altering the MAT2A active site [144], inhibiting SAM synthesis with submicromolar concentrations in vitro. However, cancer cells can adapt to PF-9366 MAT2A inhibition by upregulating MAT2A, thereby impairing the anti-proliferative effect. The inhibition of SAM synthesis likely leads to a decrease in MAT2A promoter methylation, inducing its expression. Thus, MAT2A inhibition by a small molecule may be difficult to target, unless an optimal dosage is found that impairs cancer growth and lowers the SAM levels enough without triggering an epigenetic response [144]. Fluorinated N,N-dialkylaminostilbene (FIDAS) agents were found to target the SAM-binding site of MAT2A directly and selectively [145]. FIDAS agents demonstrate more potent antitumor effects than PF-9366 and show inhibition of proliferation across a variety of cancer cell lines in vitro. In addition, the FIDAS agents can decrease tumor burden in mice [145].

## 7. Glutathione, a Crucial Metabolite against ROS

ROS are an inevitable byproduct of cellular respiration causing non-specific oxidation of lipids, nucleic acids, and proteins and subsequently inducing cell death [146]. Since ROS accumulation is a challenge for cancer cells, they induce the ROS scavenging system to increase survival. One such system is glutathione-mediated ROS deactivation. Glutathione (GSH, reduced form; GSSG, oxidized form) is a tripeptide composed of glutamate, cysteine, and glycine. It is generated by glutamate cysteine ligase (GCL) and glutathione synthase (GS) in two consecutive steps [147]. GSH is both a nucleophile and a reductant. Thus, it can react with hydrophobic electrophilic species (e.g., nitroalkene derivatives of fatty acids) to render them more hydrophilic and excretable while also being able to reduce ROS, preventing them from oxidizing macromolecules. While neutralizing ROS, GSH becomes oxidized to GSSG in the presence of GSH peroxidase (GPX) [147]. GSSG is then reduced back to GSH by the NADPH-dependent catalysis of GSH reductase (GR). Conjugation of GSH with electrophilic compounds is catalyzed by the glutathione-S-transferases (GST), a super family of Phase II detoxification enzymes [148].

Since most chemotherapeutic drugs induce ROS and GSH detoxifies it, GSH metabolism has been linked to drug resistance. GSH levels have been reported to dictate cisplatin resistance in NSCLC, ovarian, and gastric cancer cells [149,150,151]. Altering GSH levels has been shown to manipulate resistance to the epidermal growth factor receptor (EGFR) inhibitor erlotinib in EGFR-mutant NSCLC [152].

A number of agents lowering GSH levels have been designed as a means to improve the response to chemotherapy. Buthionine sulfoximine (BSO) is a synthetic amino acid which acts as an irreversible GCL inhibitor to decrease GSH synthesis. The BSO treatment sensitized multiple myeloma and neuroblastoma cancer cells to melphalan, a DNA/RNA targeting chemotherapy that interferes with the growth and spread of cancer cells [153,154]. It was also found to synergize well with cisplatin to reduce tumor burden in T47D xenografts [155]. BSO has been used in clinical trials as a part of combination therapy with melphalan in recurrent pediatric neuroblastoma (NCT00005835) [156] and pediatric neuroblastoma (NCT00002730).

GSTs are upregulated in many cancer cells including breast and colon [157,158]. As such, GST inhibition represents a promising therapeutic target, particularly of the π (pi) isoform [159,160,161]. Ethacrynic acid (EA), an FDA-approved diuretic drug and its analogue EAG (EA-2-amino-2-deoxy-d-glucose) conjugate with GSH to inhibit class-1 π 1 isoform of GST (GSTP1-1) [162]. EAG was developed to overcome the strong diuretic properties of EA. EAG was shown to have antiproliferative effects on epidermal carcinoma and leukemia cells in vitro [163]. An alkaloid, piperlongumine (PL) inhibits GSTP1-1 by forming an adduct with GSH [164] and increasing oxidative stress, and has demonstrated antiproliferative effects in human cervical and prostate cancer cells and induced cell cycle arrest [165,166,167]. PL was able to impair tumor growth and metastasis in a murine breast cancer model [168]. It enhances oral bioavailability and cytotoxicity of chemotherapeutic drugs [169]. While the drugs inhibiting GST descried above show antitumor activity, none of them exhibits GSTP1-1 specificity, which may cause off-target effects. 6-(7-nitro-2,1,3-benzoxadiazol-4-ylthio)hexanol (NBDHEX), a strong inhibitor of GSTP1-1, acts as a suicide inhibitor. NBDHEX forms an adduct with GSH in the active site of GSTP1-1 to induce inhibition [170,171]. NBDHEX is a promising therapeutic for the selective treatment of multidrug resistant tumors in leukemia and small cell lung cancer (SCLC) [172,173]. Multidrug resistant tumors often overexpress P-glycoprotein MDR1, a plasma membrane bound efflux transporter that exports drugs from the cell [174]. Interestingly, NBDHEX, not being a substrate of MDR1, is able to induce apoptosis in these resistant tumors. Good in vivo solubility and specific targeting of MDR1-expressing tumors make NBDHEX a potential treatment against multidrug resistant tumors. The isoform affinity for NBDHEX may present a challenge in clinical settings, as the class-2 µ 2 isoform of GST (GSTM2-2) is preferred to GSTP1-1 [175]. A derivative, MC3181, was developed to overcome NBDHEX’s lack of affinity to GSTP1-1. MC3181 demonstrated efficacy in mouse xenografts of vemurafenib resistant melanoma, inhibiting growth and metastasis [176]. The further development of GST isoform selective agents, such as MC3181 and NBDHEX, should enable a full accounting of their suitability to treat cancer.

## 8. Nucleotide Synthesis

### 8.1. Pyrimidine De Novo Synthesis

Amino acid catabolism provides both carbon backbone and nitrogen atoms for nucleotides. Nucleotides are categorized into either pyrimidines or purines on the basis of their nucleobase ring. Nucleotides can be newly synthesized (de novo synthesis) or can be produced from intermediates in the degradative pathway (salvage pathway). This review will focus on de novo synthesis pathway and therapeutics. The first phase of the de novo pyrimidine synthesis is catalyzed by the tri-functional multidomain enzyme, carbamoyl-phosphate synthetase 2, aspartate transcarbamylase, and dihydroorotase (CAD, also known as CPS2). First, the carbamoyl phosphate synthetase domain in CAD uses the γ-nitrogen from glutamine and bicarbonate to generate carbamoyl phosphate. The aspartate transcarbamylase (ATCase) domain in CAD then combines aspartate with carbamoyl phosphate to yield carbamoyl aspartate. Finally, dehydration mediated by the dihydroorotase (DHO) domain in CAD results in the cyclization of carbamoyl aspartate to dihydroorotate. Following the CAD sequence of reactions, the ring is oxidized into orotate by dihydroorotate dehydrogenase (DHODH), a rate-limiting enzyme in de novo pyrimidine synthesis. The orotate phosphoribosyltransferase (OPRTase) domain in uridine monophosphate synthase (UMPS) catalyzes the nucleophilic addition of orotate to phosphoribosyl pyrophosphate (PRPP) to produce orotidine-5-monophosphate (OMP). A separate domain on UMPS, OMP decarboxylase, catalyzes the decarboxylation of OMP to produce UMP. UMP can be phosphorylated to yield UDP and UTP by UMP- and UDP-kinase, respectively [177]. The remaining pyrimidines can be generated from UMP. Cytidine triphosphate (CTP) and UTP are interconverted by the CTP synthetase (CTPS), wherein γ-nitrogen of glutamine is added to and removed from CTP. TS catalyzes the methylation of dUMP to dTMP using 5,10-methyleneTHF. dTMP can be phosphorylated by deoxythymidylate kinase (DTYMK) to produce dTDP. dTDP can be phosphorylated by nucleoside diphosphate kinase (NDPK) to yield dTTP.

Cancer cells place a higher demand on nucleotide synthesis than regular cells and reprogram metabolism in ways that satisfy this requirement. Metabolic reprogramming thus creates vulnerabilities and pathway dependencies unique to the cancer cell which may be targeted. In Kirsten rat sarcoma viral oncogene homolog (KRAS)-driven, LKB1-deficient NSCLC, pyrimidine levels are maintained by upregulating mitochondrial CPS1, supplying carbamoyl phosphate to de novo pyrimidine synthesis in the cytosol [178]. The same subtype of NSCLC are also selectively sensitive to DTYMK inhibition [179]. DHODH suppression has been reported to confer a metabolic vulnerability in KRAS-driven cancers [180]. Given the importance of pyrimidine de novo synthesis in cancer, it is not surprising that several inhibitors targeting the pathway have been developed. 5-fluorouracil (5-FU) is an FDA approved uracil analog that disrupts pyrimidine homeostasis (Figure 2C and Table 2), and is used to treat many cancers, including colorectal cancer [181]. 5-fluoro-2′-deoxyuridine-5′-monophosphate (FdUMP), a molecule formed from 5-FU in vivo, inhibits TS by forming an irreversible complex with the enzyme and 5,10-methyleneTHF [181] and resulting in intracellular uracil accumulation and deoxyuridine efflux out of the cell [182]. 5-FU efficacy, however, is impaired due to rapid degradation by dihydropyrimidine dehydrogenase (DPD) [181]. Due to rapid clearance, continuous administration or high doses of 5-FU are necessary, increasing cytotoxicity. To decrease cytotoxicity, site-specific deliverable, 5-FU biodegradable carriers have been developed. Folic acid conjugated carriers containing 5-FU had a lower IC_50_ compared to 5-FU, which could enable lower dose administration [183].

Leflunomide is an FDA approved immunosuppressive agent against rheumatoid arthritis. It is a weak inhibitor of DHODH and depletes UTP and CTP ribonucleotides [184] (Figure 2C). In an in vivo AML model, it decreased the number of leukemia cells and the amount of leukemia initiating cells [185]. Leflunomide either has been used or is currently being tested in Phase 1-3 clinical trials against breast, prostate, lymphoma, brain, and multiple myeloma (NCT03709446, NCT00004071, NCT04463615, NCT00003775, NCT02509052) (Table 1). Leflunomide-mediated growth impairment of cancer cells, however, may not be directly through DHODH inhibition. Unlike other chemo drugs, leflunomide is not target-specific, which can cause potential issues. In addition to targeting DHODH, it has been shown to inhibit the PIM-1, -2, and -3 serine-threonine kinase [186]. It also has been reported to act as an agonist of the aryl hydrocarbon receptor (AhR) [187]. Another DHODH inhibitor, brequinar, demonstrated strong antitumor activity in a KRAS mutant pancreatic tumor xenograft model [180]. Despite these promising preclinical antitumor effects, brequinar failed to show efficacy in numerous clinical trials including breast [188], colon [189], and head and neck [190]. Such discrepancies may be due to rescue by the nucleotide salvage pathway. Brequinar treatment in KRAS mutant cancer cells was reversed by uridine treatment [180]. AML cells tolerate brequinar treatment due to dependence of the nucleotide salvage pathway [185]. The combination treatment of brequinar with the uridine salvage pathway inhibitors (e.g., uridine-cytidine kinase 2 (UCK2) inhibitor [191] may bolster the antitumor effect. In addition to salvage pathway inhibitors, brequinar has been tested in combination with a purine pathway inhibitor, ribavirin, in a Phase 1/2 trial (NCT03760666). Brequinar treatment in combination with purine nucleoside analogues may lead to more promising results than the treatment alone. DTYMK sits at a junction between the de novo and salvage pathways. Inhibition of DTYMK leads to dUTP incorporation into DNA during DNA double-strand break (DSB) repair and sensitizes cells to low-dose doxorubicin, a DSB-inducing chemo-drug, both in vitro and in vivo [192]. YMU1 is a mixed competitive reversible DTYMK inhibitor. It binds the catalytic site of the enzyme, competing with the substrate ATP [193]. YMU1 is an effective inhibitor, with an IC_50_ of 610nM. Upon dTTP depletion, dUTP is incorporated instead for DNA repair, causing DNA damage and cell death [192]. Efficacy of YMU1 is mostly limited to in vitro studies and thus, off target effects need further evaluation.

### 8.2. Purine De Novo Synthesis

Adenine and guanine are purine nucleobases, bicyclic structures characterized by the fusion of imidazole and pyrimidine fragments, having five nitrogen atoms. In contrast to pyrimidine biosynthesis, a ribose ring derived from PRPP, is the starting point for nucleobase construction. Amino acids and one carbon units are then used to build the imidazole ring on PRPP and subsequently annulate the pyrimidine ring onto the imidazole. Glutamine PRPP amidotransferase (PPAT) catalyzes the first committed step in de novo purine synthesis; displacing the pyrophosphate of PRPP with the γ-nitrogen of glutamine to give phosphoribosylamine. Glycine, glutamine, and aspartate are essential for construction of the purine nucleobase. The imidazole ring is constructed by (1) the donation of two carbons and one nitrogen from glycine, (2) the formylation of the glycine amino-group via 10-formylTHF, and (3) the donation of a nitrogen from glutamine. The nitrogen atom from aspartate joins to the imidazole ring and a formyl group from 10-formylTHF is added to this nitrogen atom to form a final intermediate that cyclizes with the loss of water to form inosine monophosphate (IMP). IMP is the first compound of the purine de novo synthesis pathway, and the precursor for adenosine monophosphate (AMP) and guanosine monophosphate (GMP). A fifth nitrogen from aspartate is added to generate AMP by adenylosuccinate synthetase (ASS) and adenylosuccinate lyase (ASL), two urea cycle enzymes. GMP production involves the oxidation of IMP to an intermediate (xanthosine monophosphate, XMP) by inosine monophosphate dehydrogenase (IMPDH), the rate-limiting step of de novo guanine nucleotide synthesis. De novo guanine nucleotide synthesis, specifically IMPDH, was identified as a metabolic vulnerability within a subset of SCLC characterized by low achaete-scute family BHLH transcription factor 1(ASCL1) expression [194]. Following oxidation by IMPDH, γ-nitrogen from glutamine is incorporated by GMP synthase (GMPS). A tight balance between GTP and ATP levels is maintained at the level of GMP/AMP production. Both reactions rely on the consumption of the other nucleotide species; GMP production consumes ATP and AMP production consumes GTP.

The antifolate pemetrexed inhibits tri-functional purine biosynthetic protein adenosine-3 (GART). Despite clinical efficacy, pemetrexed, along with all other antifolates, is cytotoxic due to non-specific transport by the reduced folate carrier (RFC). 2-amino-4-oxo-6-substituted pyrrolo [2¨C-d]pyrimidine derivatives are potent antifolates that inhibit GART in the nanomolar range [195]. They are poor substrates for RFC, and as a result, more effectively reduce growth of folate receptor (FR)- and proton-coupled folate transporter (PCFT)-expressing tumor cells and in human tumor xenografts [196,197]. PCFT is commonly expressed in epithelial cancers [198]. Potent inhibition of GART and target selectivity make these antifolates a promising alternative to the clinically used pemetrexed. Mycophenolic acid (MPA) is an FDA approved drug traditionally used to prevent organ rejection and was found to inhibit IMPDH (Figure 2C and Table 2). MPA has been shown to reduce the growth of a subset of SCLC-derived cell lines by depleting guanosine, GMP, and XMP [194]. Mizoribine is an imidazole nucleoside drug which has been used in clinical trials for rheumatoid arthritis, but so far has not been clinically tested in cancer. While both MPA and mizoribine are effective in vitro, mizoribine has been shown to have better biodistribution in vivo [194] and was used to access the effect of IMPDH inhibition on tumor burden in mice. Synergy with DNA synthesis targeting chemotherapeutic drug cisplatin, was also evaluated in vivo. Mizoribine alone, was less effective than cisplatin, however a combination of the two had the greatest additive effect [194]. This preclinical study suggests that targeting purine biosynthesis pathway along with other antitumor drugs can be potentially beneficial for patients with the almost-non curable disease.

## 9. Kynurenine, a Tryptophan-Derived Oncometabolite

Almost all ingested tryptophan (~99%) that is not used for protein synthesis enters the kynurenine pathway [199,200], where it is broken down to the metabolite kynurenine. Through the rate limiting enzymes indoleamine-2,3-dioxygenase 1,2 (IDO1, IDO2) and tryptophan-2,3-dioxygenase (TDO), tryptophan is converted to N’-formylkynurenine. N’-formylkynurenine then becomes kynurenine by kynurenine formamidase (AFMID) (Figure 1C). The importance of the kynurenine pathway was originally associated with its role in the biogenesis of nicotinamide adenine dinucleotide (NAD) [201]. However, the pathway has received increasing attention since it was found to be involved in many diseases including neurodegenerative diseases, inflammation, and tumor proliferation [202,203,204]. Kynurenine levels are elevated in many cancers and are correlated with poor prognosis in patients with colorectal, gynecological, melanoma, and cervical cancer [205,206,207,208,209]. Connections to tumorigenesis and, especially, the immune response render the kynurenine pathway an attractive therapeutic target.

AhR plays a central role in kynurenine-mediated impacts to cancer progression and cancer immunity. Kynurenine is an endogenous ligand of AhR, a cytosolic ligand-dependent transcription factor that is reported to promote tumorigenesis and tumor aggressiveness [210] (Figure 1C). AhR is also a receptor to the exogenous ligand 2,3,7,8-tetrachlorodibenzo-*p*-dioxin (TCDD) [211], which showed an immunosuppressive effect on T cells and dendritic cells (DCs) [212]. Increased expression of IDO1 in cancer was associated with immune escape and cancer-associated inflammation in the microenvironment [213]. Importantly, the expression of IDO1 and TDO is controlled by the AhR [214,215] and, when increased, causes depletion of local tryptophan pools in the microenvironment, suppression of antigen-specific T cell responses, and promotion of T regulatory (Treg) cell differentiation during tumor development (Figure 1C) [6]. Increased Treg cells are related to altered IDO1 expression in patients with PDAC, leukemia, and lymphoma [216,217,218]. Treg cells directly contact IDO1-expressing DCs for tolerance induction through cytotoxic T-lymphocyte-associated protein (CTLA)-4 [209,219]. Tumor repopulating cells (TRCs), a self-renewing, highly tumorigenic subpopulation of cancer cells, highly express IDO1 and transfer kynurenine to CD8+ T cells, which induces programmed cell death protein 1 (PD-1) expression through the kynurenine-AhR pathway [220].

With the growing relevance of the kynurenine pathway to various aspects of tumor biology, development of inhibitors targeting this pathway has increased with a focus on IDO1 and TDO. Several IDO1 and TDO inhibitors are in clinical trials, administered in combination with immune checkpoint inhibitors and/or chemotherapeutic drugs (Figure 1C and Table 1). Epacadostat directly binds to and inhibits IDO1 [221]. As a part of oncology combination therapy strategies, epacadostat is undergoing a Phase 1 clinical trial in rectal (NCT03516708), Phase 2 trials in pancreatic (NCT03006302), gastrointestinal (NCT03291054), lung cancers (NCT03322540), and metastatic NSCLC (NCT03322566). In Phase 3 clinical trials, epacadostat has been tested in melanoma (NCT02752074), and is being tested in urothelial (NCT03361865), and head and neck cancers (NCT03358472). BMS986205 [222], an irreversible inhibitor of IDO1, is undergoing Phase 1/2 clinical trials in liver (NCT03695250) and melanoma and lung cancer (NCT02658890). In Phase 3 clinical trials, BMS986205 is administered with an immune checkpoint inhibitor nivolumab in bladder cancer (NCT03661320) and melanoma (NCT03329846). The tryptophan analogue, indoximod, is the methylated version of tryptophan and is also an inhibitor of the IDO/TDO pathway [223]. Indoximod has been tested in Phase 2 clinical trials in melanoma with pembrolizumab or nivolumab (NCT03301636). HTI-1090 (also known as SHR-9146), a dual IDO1-TDO inhibitor, is undergoing a Phase 1 clinical trial in combination with a VEGFR2 inhibitor apatinib and the PD-1 inhibitor SHR-1210 in advanced solid tumors (NCT03491631) (Table 1). GDC-0919 is a non-competitive, dual IDO1-TDO inhibitor. It has been tested in a Phase 1 trial in advanced solid tumors, in combination with an immune checkpoint inhibitor atezolizumab (NCT02471846). Some IDO/TDO inhibitors did not pass clinical trials because of a lack of efficacy. In a 2018 Phase 3 clinical trial, a combination treatment of epacadostat and pembrolizumab failed to improve overall survival compared to the singular pembrolizumab treatment for patients with melanoma (NCT02752074). A Phase 2 trial in breast cancer was discontinued after indoximod, combined with an adenovirus-p53-transduced dendritic cell vaccine, was unable to induce adequate antitumor effects (NCT01042535) [224]. Choosing the right tumor types where the kynurenine pathway is critical for tumorigenesis, understanding the mechanism of action of inhibitors and avoiding drug combinations that target the same signaling pathway will be essential to improving treatment efficacy.

## 10. Polyamines, Arginine-Derived Oncometabolites

Polyamines are polycations that are synthesized from arginine, ornithine, and methionine. There are three polyamines-putrescine, spermidine, and spermine, and they interact with negatively charged molecules such as DNA [225], RNA [226,227,228], and proteins [229]. Through these interactions, polyamines modulate chromatin structure [230,231], alter DNA stabilization [232,233] and regulate gene expression [7].

In the first step of polyamine synthesis, arginase converts arginine to ornithine and urea (Figure 1D). Ornithine is decarboxylated to putrescine by the rate limiting enzyme ornithine decarboxylase (ODC) [8]. Putrescine is converted to spermidine by two enzymes, S-adenosylmethionine decarboxylase 1 (AMD1) and spermidine synthase (SPDSY). AMD1 provides an aminopropyl group and spermidine synthase transfers the aminopropyl group to the primary amine groups of putrescine. Similar to spermidine, the spermine synthesis requires AMD1 for an aminopropyl group synthesis and spermine synthase (SPMSY) for the aminopropyl group transferring to spermidine (Figure 1D). Spermine and spermidine can be converted back to putrescine by a two-step interconversion pathway, which involves acetylation by spermidine/spermine N^1^-acetyltransferase (SAT1) and oxidation by polyamine oxidase (PAO). Cellular demand for polyamines can also be satisfied by the polyamine transport. Although a specific polyamine transport system is currently unknown, two models have been proposed. One proposes that polyamines bind to a putative membrane transporter, undergo endocytosis, and enter the cell as an endosome-bound polyamine-receptor complex [234]. The other model suggests that, on the cell surface, polyamines bind to the heparan sulfate side chains of recycling glypican 1 and are internalized by endocytosis [235]. Recently, CD98 (4F2 heavy chain antigen, encoded by *SLC3A2*) has been identified as a key polyamine transporter in neuroblastoma [236].

Polyamine metabolism is regulated by various oncogenes including Myc and RAS. Elevation of Myc, which regulates the transcriptional activity of ODC, has been reported to increase both ODC mRNA and protein levels in neural and breast cancer cells [237,238,239]. The PTEN-PI3K-mTORC1 pathway is linked to polyamine metabolism in prostate cancer via AMD1 upregulation. mTORC1 induces AMD1 expression, stabilizing the AMD1 proenzyme by phosphorylating Ser298 [240]. Increased polyamine transport activity has been reported to be linked to BRAF inhibitor resistance in BRAF-mutated melanomas [241]. Elevated polyamine levels and those of related metabolites (e.g., N-acetylputrescine, cadaverine, and 1,3-diaminopropane) can be used as a biomarker for cancer. Intermediary metabolites in polyamine metabolism may be a useful diagnostic in lung and liver cancer [242,243]. High serum levels of spermidine are closely associated with poor prognosis in HER2 positive breast cancer patients treated with trastuzumab [244]. The research collectively implies that altered polyamine metabolism in cancer is an attractive target for therapeutic intervention. So far, several inhibitors targeting polyamine metabolism have been developed. Difluoromethylornithine (DFMO) is an inhibitor of ODC (Figure 1C and Table 2). It was originally approved by the FDA for the treatment of *Trypanosoma brucei* infections, and later found to irreversibly inhibit ODC. DFMO treatment induced depletion of putrescine and spermidine in vitro, causing p27-RB coupled G1 cell cycle arrest [238,245]. A combination treatment of DFMO and PCV (Nitrosourea-based multidrug chemotherapy; procarbazine, lomustine, and vincristine) was administered in a Phase 3 clinical trial for GBM and astrocytoma. While this therapeutic regimen enhanced patient survival in astrocytoma, it did not improve prognosis in GBM [246,247].

DFMO monotherapy can increase polyamine transport when intracellular polyamine levels are reduced. To circumvent this issue, a polyamine transport inhibitor, such as Trimer44NMe and lipophilic lysine-spermine conjugates, can be combined with DFMO. In skin cancer and a gemcitabine-resistant pancreatic cancer model, a combination treatment caused polyamine depletion and tumor growth reduction [248,249,250]. The combination treatment of DFMO and AMXT1501, a polyamine transport inhibitor, is in a Phase 1 clinical trial for advanced solid tumors (NCT03536728) (Figure 1D and Table 1). Methylglyoxal bis(guanylhydrazone) (MGBG) [251,252], 4-amidinoindan-1-one 2′-amidinohydrazone (SAM486A) [253], 5′-(((Z)-4-amino-2-butenyl)methylamino)-5′-deoxyadenosine (AbeAdo) [254], and its 8-methyl derivative (Genz-644131) [255] are irreversible inhibitors of AMD1 and impede polyamine synthesis. The efficacy of MGBG and SAM486A was tested in Phase 2 clinical trials, but discontinued due to toxicity [253]. Further chemical optimization and exploration of the combination treatment with other anti-cancer drugs has been done to address toxicity and show improvements in vivo [256,257]. Although preliminary, these preclinical study results are encouraging and suggest that the AMD1 inhibitors may move towards clinical trials.

The combination treatment of DFMO with SAM486A has been shown to suppress tumor growth in transgenic and xenograft neuroblastoma mouse models [258]. Polyamine analogues can also be used as an antitumor therapy. For example, N1, N11-bis(ethyl) norspermine (BENSpm/DENSpm), N1-ethyl-N11-((cyclopropyl)methyl)-4,8-diazaundecane (CPENSpm), PG-11093, and PG-11047 downregulate the biosynthesis and upregulate the catabolism of polyamines without significant toxicity. They reduce cancer cell growth when used as a monotherapy or co-treatment with chemotherapeutic drugs [259,260]. While promising, the polyamine analogues need further optimization to move forward. For example, DENSpm has been tested in clinical trials for hepatocellular carcinoma [261], metastatic breast cancer [262], and lung cancer [263]. Unfortunately, the single agent was unable to reduce tumor burden. These results suggest that the combination treatment with other anti-cancer drugs may improve therapeutic efficacy.

## 11. Conclusions

Since the development and subsequent clinical success of the folate analogue aminopterin and uracil analogue 5-FU in treating cancer, significant investments from both academia and industry have been made to discover drugs targeting other amino acid-related pathways for clinical use. Despite these efforts, the number of novel FDA-approved drugs targeting amino acid metabolism has scarcely grown over the past few decades. One of the main reasons for the stagnant progress is the lack of in vivo efficacy due to metabolic redundancy or metabolic compensation. Targeting AATs is challenging because multiple AATs can transport the same amino acid. Moreover, some cancer cells can circumvent transporter system blockade by obtaining amino acids from the microenvironment through macropinocytosis. Metabolic compensation also needs to be considered when targeting biosynthesis pathways. Cancer cells upregulate serine biosynthesis, but their serine uptake is also enhanced compared with normal tissues. To make PHGDH inhibitors work in vivo, circulating serine levels should be lowered (e.g., serine-deficient diet), however this is not practical in clinical settings. Likewise, cancer cells often increase nucleotide salvage pathway utilization, which mitigates cytotoxic effects of drugs targeting the nucleotide synthesis pathway. Therefore, a more effective oncology therapeutic strategy should include targeting multiple metabolic pathways simultaneously (e.g., targeting both nucleotide synthesis pathway and the salvage pathway) or targeting a specific amino acid metabolic pathway with therapies against oncogenic signaling pathways (e.g., RTK inhibitors). Recent combination treatments of drugs targeting amino acid metabolism and chemotherapeutic drugs are undergoing clinical trials in various cancers and the preliminary results are encouraging.

Another aspect to consider is the metabolic interaction between cancer cells and immune cells. For example, the important role of kynurenine in cancer aggressiveness is linked to its ability to suppress antitumor immune responses. The efficacy of IDO1/TDO inhibitors can be fortified by therapies modulating immune cell function. Indeed, IDO/TDO inhibitor monotherapy failed to show in vivo efficacy and all the current clinical trials of IDO/TDO inhibitors are combination therapies with immune checkpoint inhibitors (Table 1). As more novel therapeutic regimens targeting both amino acid-associated metabolism and non-metabolic signaling pathways move towards the clinic, it will be thrilling to see how the final clinical results turn out. If approved, it would be a breakthrough in the field of cancer metabolism and provide encouragement for further metabolism-centric drug development.

## Figures and Tables

**Figure 1 cells-09-01904-f001:**
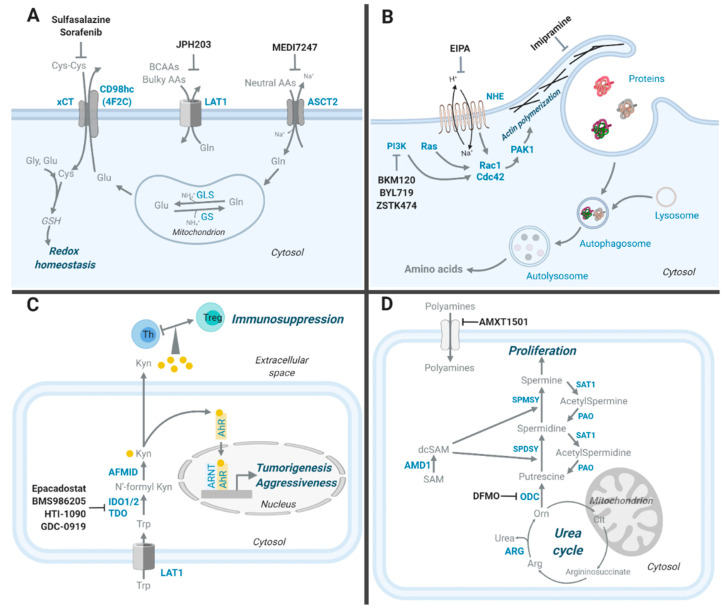
Amino acid metabolic pathways altered in cancer and select inhibitors targeting these pathways. (**A**) Three main amino acid transporters (AATs) upregulated in cancer cells and transporter-targeting inhibitors that are either being used clinically or tested in clinical trials are shown. (**B**) A simplified macropinocytosis process is depicted and inhibitors targeting the process are shown. Rat sarcoma virus oncogene homolog (Ras) and phosphatidylinositol-3-kinases (PI3Ks) positively regulate macropinocytosis by promoting Rac1 and Cdc42-mediated actin polymerization. (**C**) Kynurenine is a catabolite derived from tryptophan. Key enzymes in the kynurenine synthesis pathway, regulatory mechanisms of kynurenine, and IDO/TDO inhibitors tested in clinical trials are shown. (**D**) Polyamines are catabolites derived from arginine. Polyamine synthesis and catabolic pathways are depicted, and polyamine metabolism inhibitors tested in clinical trials are shown. Metabolites are in grey, proteins are in blue, and inhibitors are in black. Proteins: ASCT2: Alanine-serine-cysteine transporters 2; LAT1: L-type amino acid transporter 1; xCT/CD98hc: 4FC2 cystine/glutamate heterodimeric antiporter; GLS: Glutaminase; GS: Glutamine synthetase; NHE: Na^+^/H^+^ exchanger; Rac1: Ras-related C3 botulinum toxin substrate 1; Cdc42: Cell division control protein 42; PAK1: P21-activate kinase 1; PI3K: Phosphoinositide 3-kinases; IDO1/2: Indoleamine-2,3-dioxygenase 1,2; TDO: Tryptophan-2,3-dioxygenase; AFMID: Kynurenine formamidase; AhR: Aryl hydrocarbon receptor; ARNT: Aryl hydrocarbon receptor nuclear translocator; ARG: Arginase; ODC: Ornithine decarboxylase; AMD1: S-adenosylmethionine decarboxylase 1; SPDSY: Spermidine synthase; SPMSY: Spermine synthase; SAT1: Spermidine/spermine N^1^-acetyltransferase; PAO: Polyamine oxidase. Metabolites: Cys-Cys: Cystine; Cys: Cysteine; Gly: Glycine; Glu: Glutamate; Gln: Glutamine; Neutral AAs: Neutral amino acids (alanine, serine, cysteine, glutamine); BCAAs: Branched-chain amino acid (valine, leucine, isoleucine); bulky AAs: Bulky amino acids (phenylalanine, methionine, histidine, tryptophan, tyrosine); Trp: Tryptophan; N’-formylkyn: N’-formylkynurenine; Kyn: Kynurenine; Cit: Citrulline; Arg: Arginine; Orn: Ornithine; SAM: S-adenosylmethionine; dcSAM: Decarboxylated. Cells: Th: T-helper cell; Treg: Regulatory T cell. Inhibitors: DFMO: Difluoromethylornithine.

**Figure 2 cells-09-01904-f002:**
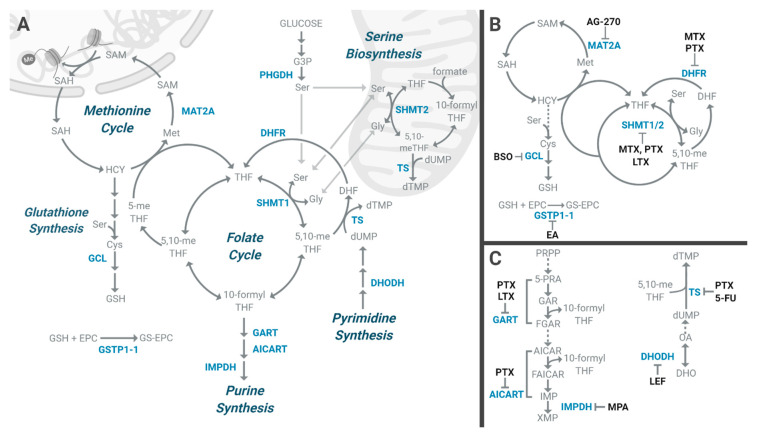
Targetable vulnerabilities related to serine and 1C metabolism. (**A**) Metabolic pathways linked to serine, glycine, and 1C metabolism are shown. Serine biosynthesis requires nitrogen from glutamate and three carbons from the glycolytic intermediate G3P. Serine metabolism is directly linked to the folate cycle, which transfers one carbon unit between metabolites using THF as a carrier. In the mitochondria, serine catabolism produces glycine and 5,10-meTHF in the cytosol 5,10-meTHF is used to synthesize serine from glycine. In both compartments, 5,10-meTHF supports dTMP synthesis from dUMP, a downstream metabolite of pyrimidine synthesis. In the process 5,10-meTHF is converted to the non-carrier DHF, which is converted to THF to re-enter the folate cycle. Purine synthesis is also connected to the folate pathway through a one carbon transfer from 10-formyl. The most reduced carrier is 5-meTHF, which is derived from 5,10-meTHF. The oxidation of 5-meTHF to THF biologically functions to regenerate Met from HCY in the methionine cycle. This ensures a stable pool of SAM for DNA and histone methylation. Through HCY the methionine cycle is connected to the transsulfuration pathway, which generates GSH for ROS mitigation. (**B**,**C**) Inhibitors targeting enzymes in the folate cycle, the methionine cycle and glutathione synthesis are shown in (**B**) and inhibitors targeting nucleotide metabolism are illustrated in **C**. Metabolites are in grey, proteins are in blue, and inhibitors are in black. Proteins: AICART: 5-aminoimidazole-4-carboxamide ribonucleotide formyltransferase; DHFR: Dihydrofolate reductase; DHODH: Dihydroorotate dehydrogenase; GART: Phosphoribosylglycinamide formyltransferase; GCL: Glutamate-cysteine ligase; GSTP1-1: Class-1 π 1 isoform of glutathione s-transferase; IMPDH: Inosine monophosphate dehydrogenase; MAT2A: Methionine adenosyltransferase 2A; PHGDH: Phosphoglycerate dehydrogenase; SHMT1: Serine hydroxymethyltransferase 1; SHMT2: Serine hydroxymethyltransferase 2; TS: Thymidylate synthetase. Inhibitors: PTX: Pemetrexed; MTX: Methotrexate; LEF: Leflunomide; LTX: Lometrexol; M/P/LTX: Methotrexate, Pemetrexed, and Lometrexol; BSO: Buthionine sulphoximine; EA: Ethacrynic acid; MPA: Mycophenolic acid; AG-270; 5-FU: 5-flurouracil. Metabolites: SAM: S-adenosyl methionine; SAH: S-adenosyl-l-homocysteine; HCY: Homocysteine; Cys: Cysteine; GSH: Reduced glutathione; THF: Tetrahydrofolate; 5-meTHF: 5-methylTHF; 5,10-meTHF: 5,10-methyleneTHF; 10-formylTHF: 10-formylTHF; DHF: Dihydrofolate; Ser: Serine; Gly: Glycine; dTMP: Deoxythymidine monophosphate; dUMP: Deoxyuridine monophosphate; 5-PRA: Phosphoribosylamine; GAR: Glycineamide ribonucleotide; FGAR: N-formylglycinamide ribonucleotide; AICAR: 5-aminoimidazole-4-carboxamide ribonucleotide; FAICAR: 5-formamidoimidazole-4-carboxamide ribotide; IMP: Inosine monophosphate; XMP: Xanthosine 5′-monophosphate; DHO: Dihydroorotate; OA: Orotate; G3P: Glyceraldehyde 3-phosphate.

**Table 1 cells-09-01904-t001:** Cancer therapeutics in clinical trials targeting amino acid metabolism.

Pathway	Target	Inhibitor	Phase	Trial ID	Target Cancer	Last Updated Dates, Status
Amino acid transporter	ASCT2	MEDI7247	Phase I	NCT03106428	Hematological cancers	Completed
	LAT1	JPH203	Phase II	UMIN000034080	Advanced biliary tract cancer	Completed
Macropinocytosis	PI3K	BKM120	Phase II	NCT02301364	Lymphoma	Completed
				NCT01297491	Lung cancer	Completed
			Phase III	NCT01610284	Breast cancer	Completed
		ZSTK474	Phase I	NCT01280487	Advanced solid tumors	Completed
Polyamines	ODC and polyamine transport	DFMO with AMXT1501	Phase I	NCT03536728	Advanced solid tumors	13 May 2020 Recruiting
Serine-folate cycle	GART; SHMT1/2	Lometrexol with paclitaxel	Phase I	NCT00024310	Advanced solid tumors	17 September 2013
		Lometrexol	Phase II	NCT00033722	Stage IIIB and IV NSCLC	6 January 2014
Methionine cycle	MAT2A	AG-270 with docetaxel, nab-paclitaxel, and gemcitabine	Phase I	NCT03435250	Advanced solid tumor or lymphoma with homozygous MTAP deletion	10 July 2020 Recruiting
Glutathione	GCL	Buthionine sulfoximine (BSO) with Melphalan	Phase I	NCT03435250	Neuroblastoma in pediatric patients	1 February 2017
		BSO with Melphalan (followed by bone marrow or peripheral stem cell transplantation)	Phase I	NCT00002730	Resistant or recurring neuroblastoma in pediatric patients	30 August 2016
Kynurenine	IDO1	Epacadostat + chemoradiation	Phase I	NCT03516708	Rectal cancer	18 June 2020 Recruiting
		Epacadostat+Pembrolizumab+CRS-207+/- Cyclophosphamide/GVAX	Phase II	NCT03006302	Pancreas cancer	12 February 2020 Recruiting
		Epacadostat+Pembrolizumab	Phase II	NCT03291054	Gastrointestinal cancer	18 December 2019 Active
		Pembrolizumab +/- Epacadostat	Phase II	NCT03322540	Lung cancer	5 February 2020 Active
		Pembrolizumab+ platimun based therapy (pemetrexed, carboplatin, cisplatin or paclitaxel) +/- Epacadostat	Phase II	NCT03322566	Metastatic NSCLC	29 January 2020 Active
		Pembrolizumab +/- Epacadostat	Phase III	NCT02752074	Melanoma	Completed
		Pembrolizumab +/- Epacadostat	Phase III	NCT03361865	Urothelial cancer	Active
		Pembrolizumab +/- Epacadostat	Phase III	NCT03358472	Head and neck cancer	Active
		BMS986205 with Atezolizumab	Phase I	NCT02471846	Advanced solid tumors	Completed
		BMS986205 with Nivolumab	Phase I/II	NCT03695250	Liver cancer	25 February 2020 Recruiting
		BMS986205 with Nivolumab+ Ipilimumab	Phase I/II	NCT02658890	Lung cancer and Melanoma	9 June 2020 Recruiting
		BMS986205 with Nivolumab+Gemcitabine+Cisplatin	Phase III	NCT03661320	Bladder cancer	29 June 2020 Recruiting
		BMS986205 with Nivolumab	Phase III	NCT03329846	Melanoma	16 June 2020 Active
	IDO1/TDO	HTI-1090/SHR-9146 with SHR-1210 and Apatinib	Phase I	NCT03491631	Advanced solid tumors	6 June 2018 Unknown
Pyrimidine synthesis	DHODH	Leflunomide	Phase I/II	NCT03709446	Metastatic Triple Negative Breast cancer	17 March 2020 Recruiting
		Mitoxantrone and Prednisone +/- Leflunomide	Phase II/III	NCT00004071	Prostate Cancer	11 September 2012
		Leflunomide	Phase II	NCT04463615	Recurrent and refractory lymphoproliferative disorders	9 July 2020 Not yet recruiting
		Leflunomide	Phase II	NCT00003775	Brain and central nervous system tumor	13 September 2012
		Leflunomide	Phase I/II	NCT02509052	Recurrent and refractory plasma cell myeloma	6 September 2019 Active

**Table 2 cells-09-01904-t002:** FDA approved cancer therapeutics targeting amino acid metabolism.

Pathway	Inhibitor	FDA Appoved for	Original Target	Metabolic Target
Amno acid transporter	Sulfasalazine	Anti-inflammatory	Unclear	xCT
	Sorafenib	Cancer (Liver; kidney; thyroid)	RTK	xCT
Macropinocytosis	Imipramine	Anti-depressant	Tricyclic antidepressants	Membrane ruffle formation
	BYL719	Cancer (HR+/HER2- advanced breast cancer)	PI3K	Actin polymerization for membran ruffle formation
Folate cycle	Pemextred	Cancer (Non-small cell lung cancer; pleural mesothelioma)	DHFR; TS; AICART;GART	DHFR; TS; AICART;GART
	Methotrexate	Cancer (Leukemia; breast cancer; lymphoma; osteosarcoma; primary central nervous system lymphoma)	DHFR	DHFR
Glutathione	Ethacrynic Acid (EA)	Diuretic agent	Na-K-Cl cotransporter in the thick ascending loop of Henle and the macula densa	GSTP1-1
Nucleotide synthesis	5-Flurouracil (5-FU)	Anti-neoplastic agent; uracil analog	TS	TS
	Leflunomide	Immunosuppressive agent	DHODH	DHODH
	Mycophenolic Acid (MPA)	Immunosuppressive agent to prevent organ rejection	IMPDH	IMPDH
Polyamine	Difluoromethylornithine(DFMO)	*Trypansoma brucei infection*	ODC	ODC

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
