# Peer review of "Oncology Therapeutics Targeting the Metabolism of Amino Acids"

_cells, 2020, doi:10.3390/cells9081904_

Round 1
Reviewer 1 Report
This review is well-written and articulated. It encompasses a thorough review of the literature dealing with the original studies in the area of cancer metabolism with focus on the metabolism of amino acids. This review deals with an area of research (amino acid metabolism in cancer) that is of upmost interest for many researchers and clinicians, so in my opinion it could facilitate the apprehension of the great amount of information that has emerged from the mechanistic studies and clinical drug development and the utilization of such information to explore the extent to which the metabolism of amino acids is involved in a particular type of cancer and identify the optimal target for therapeutic perturbation.
I do not have any further comments to the review, and I think it could be disseminated in its actual form.
Author Response
There were no critiques from this reviewer to address, so our revision focused on making sure the paper complied with editorial guidelines.
Reviewer 2 Report
In the manuscript “Oncology Therapeutics Targeting the Metabolism of Amino Acids”, the authors summarize and discuss how cancer cells acquire and use amino acids, delineate the targets related to amino acid metabolism and promising therapeutic approaches. The review is interesting and well written; I have only minor suggestions:
- Please increase the resolution of the Figure 1 and Tables 1-2.
- Please explain the abbreviations in the text:
Line 3—NEAA; Line 68-FDA; Line 95- Ras, PI3K; Line 128-mTORC1; Line 201-PTEN; Line 294-PHGDH; Line 295-3PG, PHP; Line 414-PDX; Line 466-MAT; Line 474-NSCLC; Line 475-GBM; Line 575-KRAS; Line 578-DTYMK, DHODH; Line 622-PRPP; Line 676-AhR.
- In Referenses:
Please provide pages for references nr: 10, 31, 75, 149, 236, 241, 243, 248.
Please provide Volume for references nr: 60, 149.
- Line 104 and 106- Please use normal font (not italic) for words “protein”, “aryl hydrocarbon receptor”.
- Line 171- VDAC2 (without bold).
Author Response
We thank the Referee for their detailed analysis of the paper. Each comment from the Referee #2 is addressed below.
- Please increase the resolution of the Figure 1 and Table 1-2: We increased the size of the Figure 1 and edited Tables 1-2.
- Please explain abbreviation in the text: We included FDA, Ras, PI3K, PTEN, but the rest of the abbreviation has already been in the text (NEAA: abstract, PHGDH:252, PHP:251, 3PG:252, PDX:86, MAT:439, NSCLC:170, GBM:263, DTYMK:572, DHODH:563 (main text) and 345 (Figure legend),PRPP:566, AhR:603)
- References: i) Please provide pages for references: We included page information. When page information is not provided by the journal, doi information is included instead. ii) Please provide volume for references nr 60, 149: ref 60 is a book chapter. Thus we included edition # in lieu of volume. For 149, volume is 2010, which had been provided in the initial manuscript.
- Line 104 and 106: Please use normal font for words "protein", "aryl hydrocarbon receptor": It has been fixed.
- Line 171-VDAC2 (without bold): It has been fixed.